# Designing a Microfluidic Chip Driven by Carbon Dioxide for Separation and Detection of Particulate Matter

**DOI:** 10.3390/mi14010183

**Published:** 2023-01-11

**Authors:** Ruofei Wang, Heng Zhao, Xingbo Wang, Jiaqi Li

**Affiliations:** Centre for Lidar Remote Sensing Research, School of Mechanical and Precision Instrument Engineering, Xi’an University of Technology, Xi’an 710048, China

**Keywords:** virtual impactor, computational fluid dynamics, microfluidic chip, dynamic viscosity, particulate matter, carbon dioxide

## Abstract

Atmospheric particulate pollution poses a great danger to the environment and human health, and there is a strong need to develop equipment for collecting and separating particulate matter of different particle sizes to study the effects of particulate matter on human health. A virtual impactor is a particle separation device based on the principle of inertial separation which provides scientific guidance for identifying the composition characteristics of particles. Much existing virtual impactor research focuses on the design of structural dimensions with little exploration of the effect of fluid properties on performance. In this paper, a microfluidic chip with a cutoff diameter of 1.85 µm was designed based on computational fluid dynamics and numerically simulated via finite element analysis to analyze important parameters such as inlet flow rate, splitting ratio and fluid properties. By numerical simulation of the split ratio, we found that the obtained collection efficiency curves could not be combined into one characteristic curve by the ${\mathrm{Stk}}^{0.5}$ scaling method. We therefore propose a modified Stokes number equation for predicting the cutoff diameter at different splitting ratios. The collection efficiency curves of different fluids as microfluidic chip media were plotted, and the results show that the cut particle size was reduced from 2.5 µm to 1.85 µm after replacing conventional fluid air with CO_2_ formed by dry ice sublimation. This is a decrease of approximately 26%, which is superior to other existing methods for reducing the cutoff diameter.

## 1. Introduction

Particulate matter (PM) is both a major driver of climate change and a source of toxicity to human health. Various studies have shown that chronic exposure to particulate matter has multiple health effects on people of all ages [1,2,3]. Studies on PM_2.5_ have shown that high concentrations of particulate matter seriously affect the quality of weather, creating bad weather such as haze, which poses a great danger to human health [4]. There is a strong need to develop devices for collecting and separating particles of different particle sizes to study the effects of particles on human health. A virtual impactor is a device used for the inertial separation of particles in the air. In a virtual impactor, when the airflow changes the flow direction, the particles with less inertia inside it follow the airflow, while the particles with more inertia escape from the airflow, thus achieving a separation device based on particle size [5]. Since its development in 1966, the virtual impactor has been found to have properties missing in conventional inertial impactors, i.e., the particles are collected in the probe to avoid particle bounce effects [6]. Since then, most researchers have investigated the properties of virtual impactors by means of finite element analysis. In 1974, Marple investigated the effects of jet-to-plate distance, the jet Reynolds number and jet throat length on the efficiency characteristic curves of the impactor by means of finite element analysis [7]. In 1980, Marple determined the properties of a virtual impactor through the numerical solution of the Navier–Stokes equations and the particle equations of motion. The effects of the parameters of virtual impactors on collection efficiency and wall loss have also been investigated [8].

Most of the existing research on virtual impactors focuses on the following points: (1) reducing the cutoff diameter of virtual impactors without increasing the pressure drop; (2) developing virtual impactors with low cutting points; (3) reducing the wall loss without affecting the collection efficiency of virtual impactors; (4) reducing the pressure drop of virtual impactors at the same flow rate, and (5) the integration of virtual impactors with other sensors. For example, in 2006, Prachi Middha proposed a novel approach to the design of virtual impactors for nanoparticle slots. The compressible flow through the slot impactor was simulated using a computational fluid dynamics method and the corresponding particle trajectories were investigated. The results of the study showed that the developed design can be sized by simply varying the working pressure of the instrument. With a 20-fold increase in operating pressure, the 50% cutoff diameter was varied from 13 nm to 200 nm without significantly affecting the small flow ratio to the total flow ratio [9]. In 2009, Yong-Ho Kim et al. were able to accelerate particles smaller than 100 nm by applying a DC potential to an integrated electrode pair. Through electrical acceleration, the large pressure drop can be significantly reduced and a cutoff diameter of less than 100 nm can be successfully achieved [10]. In 2013, Handol Lee investigated the effect of orifice plates on the collection efficiency and wall loss of virtual impactors through numerical simulation and experimental validation, and the results showed that orifices are beneficial to reducing the cutoff diameter of slit virtual impactors and the wall loss of the collection nozzle. However, the orifice plate caused some wall loss at the inlet, and further studies are needed to solve this problem [11]. In 2016, Hsiao-Lin Huang investigated the effect of sheath airflow on virtual impactor characteristics via finite element analysis, and the results showed that sheath airflow can maintain the particle flow in the middle of the flow channel with no loss to the wall of the virtual impactor [12]. In 2018, Muhammad Zeeshan Zahir conducted a numerical and experimental study on two slit nozzle virtual impactors, one with a three-compartment horizontal inlet and the other with a typical vertical inlet. The results showed that the cutoff diameter was reduced from ${\mathrm{Stk}}^{0.5}=0.80$ to ${\mathrm{Stk}}^{0.5}=0.41$ and the wall loss near the cutoff diameter was reduced from 25–45% to less than 4.5% using a three-compartment horizontal inlet. In addition, the use of a three-compartment horizontal inlet reduced contamination from particles smaller than the cutoff diameter of the virtual impactor to nearly zero [13]. Muhammad Zeeshan Zahir developed a dual partitioned horizontal inlet in order to improve the collection efficiency of the slit virtual impactor and to reduce wall loss problems. These two partitions are provided to supply both aerosol and clean air to the virtual impactor. The developed inlet configuration was investigated numerically and experimentally by considering different flow ratios of aerosol to clean air. The results of the study showed cutoff diameters of PM_2.5_, PM_5_ and PM_10_ virtual impactors were reduced by 16, 10 and 11%, respectively, while the wall loss of the particle cutoff diameter was reduced from 16% to about 1% in all three cases [14]. In 2019, Ta-Chih Hsiao investigated the flow field and particle trajectories inside a linear slit-type virtual shocker via finite element analysis, and proposed a new virtual shocker modified Stokes number (${\mathrm{Stk}}_{c}{}^{0.5}$) to predict the $d_{50}$ for different values of the splitting ratio [15]. In 2022, Heng Zhao investigated the effect of temperature on the collection efficiency of virtual impactors through the relationship between temperature and hydrodynamic viscosity, and the results showed that the cutoff diameter of virtual impactors could be effectively reduced by reducing the temperature without increasing the pressure drop [16].

In the study of virtual impactor and sensor integration, virtual impactors are mostly applied to particulate matter sensors. Having good separation and collection performance, virtual impactors provide powerful assistance for the subsequent study of particulate matter. In 2007, Yong-Ho Kim et al. reported an integrated air particle classification module consisting of a micromachined three-stage virtual impactor for classifying air particles according to their size and a flow distributor for supplying the required flow to the virtual impactor. Experimental results showed that the three-stage virtual impactor had cutoff diameters of 135 nm, 1.9 µm and 4.8 µm, respectively [17]. In 2008, Yong-Ho Kim et al. reported an integrated particle detection chip for low-cost environmental monitoring that consisted of a micro-virtual impactor and a micro-corona discharger. Experimental results showed that their designed integrated chip had a cutoff diameter of 550 nm or 1.1 mm [18]. In 2017, Mingzhi Dong et al. presented the design of a PM_2.5_ sensor with VI functionality optimized for integrated VI through simulation-assisted analysis and implemented the sensor using silicon micromachining. The results of the study demonstrated the feasibility of their PM_2.5_ sensor design concept, which had an accuracy of up to $2.55\;\mathrm{ug}/m^{3}$ [19]. In 2019, Jin Xie et al. proposed a PM1 aerosol sensor with a virtual impactor integrated with an indicated acoustic wave (SAW) sensor. The virtual impactor was fabricated using 3D printing technology for PM1 separation. Its measurement accuracy was $7.446\;\mathrm{Hz}/\min\mathrm{per}\;\mathrm{ug}/m^{3}$ [20]. Ning Xue et al. designed a two-stage PM air-microfluidic graded chip with cutoff diameters of 10 µm and 2.5 µm based on COMSOL numerical simulations, and numerically analyzed the virtual impactor parameters [21]. In 2022, Jianhai Sun et al. designed a two-stage air microfluidic circuit and explored the effects of various factors on collection efficiency and wall loss through numerical simulations, optimizing the design of the microfluidic circuit structure [22]. Most studies on virtual impactors affect the cutoff diameter and wall loss of virtual impactors by changing the size of the virtual impactor or the inlet flow rate; however, few studies have investigated the effect of fluid dynamic viscosity on the cutoff diameter. According to the Stokes number equation, the dynamic viscosity is a key factor affecting the cutoff diameter of the virtual impactor, and it is also related to temperature and fluid properties, which have been little studied so far.

In this paper, we focus on the effect of temperature and fluid properties on the cutoff diameter of the virtual impactor. Based on this we propose a new method to reduce the virtual impactor cutoff diameter using CO_2_ as a fluid based on the Stokes number and numerical simulation. This method achieves the reduction of the cutoff diameter by changing the fluid dynamic viscosity and temperature inside the virtual impactor. The method proposed in this paper uses CO_2_ as a fluid to reduce the size of the cut particles by sublimating dry ice to form cryogenic CO_2_, thus combining the advantages of both cryogenics and CO_2_. Numerical simulation results show that the effect of using CO_2_ as a fluid to reduce the cutoff diameter is similar to that of introducing sheath gas at room temperature; in cryogenic environments, the effect of this method of reducing the cutoff diameter to 2.5 µm is far superior to other current methods. The advantage of this method is that gaseous CO_2_ formed by the sublimation of dry ice can be used as a fluid, and it is possible to obtain not only the low dynamic viscosity of CO_2_ itself but also to take advantage of the cryogenic environment created by its sublimation. The combination of cryogenics and CO_2_ gas together reduces the cutoff diameter of VI, and the resulting effect is better than the existing common methods. In addition, we also explore the effect of inlet flow (Q), splitting ratio (r) and other factors on the virtual impactor. In the study of the splitting ratio, we find that when the splitting ratio (r) of the virtual impactor is varied, the obtained collection efficiency curves cannot be combined into one characteristic curve via ${\mathrm{Stk}}^{0.5}$ scaling. This suggests that it is not possible to predict the cutoff diameter at different splitting ratios from the Stokes number. Therefore, we propose a modified Stokes number equation (${\mathrm{Stk}}_{m}$) for predicting the cutoff diameter at different splitting ratios. The collection efficiency curves obtained from ${\mathrm{Stk}}_{m}$ can be combined into one characteristic curve, indicating that ${\mathrm{Stk}}_{m}$ is able to predict the cut particle size at different splitting ratios. Detailed theoretical analysis and numerical simulations are given in Section 2 and Section 3. We finally obtain a virtual impactor cutoff diameter of 1.85 µm and a good steepness of the collection efficiency curve.

## 2. Theoretical Analysis and Methods

### 2.1. Theoretical Analysis

When particles follow the airflow inside the virtual impactor (VI), the trajectory of the particles is related to their particle size. Fine particles follow the flow line of the airflow, while coarse particles keep a straight line motion due to their own inertia [23]. Figure 1 shows the particle flow line distribution of the VI. 

The particles enter the interior of the VI through an accelerating jet (I), the fine particles follow the fluid motion to the major flow outlet (II), and the relatively large particles maintain a linear motion due to their own inertia to the minor flow outlet (III). The acceleration nozzle width (W), the minor flow channel width (L), and the distance from the jet to the plate (S) are the key parameters for designing the VI. Since the VI can be used as an aerosol particle separator, its performance is evaluated using the collection efficiency (CE) and wall loss (WL). CE is defined as the ratio of the number of particles exiting the major flow to the sum of the number of particles exiting the major flow and the minor flow, and WL is defined as the ratio of the number of particles lost inside the VI to the number of inlet particles [15]. The equations for CE and WL are shown below.
(1)$$\mathrm{CE}=\frac{N_{\mathrm{major}}}{N_{\mathrm{major}}+N_{\mathrm{minor}}}$$
(2)$$\mathrm{WL}=\frac{N_{\mathrm{in}}-N_{\mathrm{major}}-N_{\mathrm{minor}}}{N_{\mathrm{in}}}$$

The ideal separation curve for the VI as an inertial separation device for particles is a step function. This means that when the size of the particles is larger than the cutoff diameter, they will escape from the flow line and enter the minor flow channel. When the size of the particles is smaller than the cutoff diameter, they will follow the streamline and reach the exit of the major flow channel. In the VI design, the particle Stokes number is defined as the ratio of the particle stopping distance to the characteristic length (a dimensionless parameter) and is used as an indicator of particle behavior in the virtual impactor [24].
(3)$$\mathrm{Stk}=\frac{\rho_{p}d_{p}{\mathrm{QC}}_{c}}{9{\mu W}^{2}H}$$

Here, Q is the nozzle inlet flow rate, $C_{c}$ is the Cunningham correction factor, $\rho_{p}$ is the particle density and $d_{p}$ is the particle size.

Particles with the same Stokes number exhibit similar trajectories during inertial separation. In a real VI, its CE curve cannot exhibit the ideal state, so the steepness of the CE curve is also an important factor to be considered in VI design.

### 2.2. Methods

We performed numerical simulations using COMSOL Multiphysics software, which accurately predicts the particle trajectories using finite element analysis. In this paper, we investigated the results of particle trajectories for different particle sizes as well as the CE of the VI at different temperatures and with different fluids. In the numerical simulation, we first verified the grid independence and inlet particle number independence, then we selected the grid number and inlet particle number that could effectively represent the numerical simulation results while reducing the complexity of the model and improving the computing time. At the same time, we used a two-dimensional model to simulate the laminar flow and particle trajectory tracking. Compared with the 3D model, the number of meshes and operations were greatly reduced, and computation time was saved while the simulation results remained basically the same.

To make the cutoff diameter of the VI 2.5 µm, we first set $d_{50}$ to 2.5 µm while temporarily setting the nozzle inlet width (W) to 1.5 mm. For the shape of the nozzle, a previous study showed that rectangular versus circular nozzles do not have a significant effect on virtual impactor performance. However, the rectangular nozzle is subjected to the least pressure at the same flow rate. Therefore, we set the nozzle inlet as rectangular (W = H) and also selected the recommended ${\mathrm{Stk}}_{50}$ of 0.59 [19]. The Cunningham correction factor was calculated to be 1.0665. The inlet flow rate Q of the nozzle was calculated to be 2.9 L/min, and the nozzle inlet Reynolds number Re was derived to be 2159, which belongs to the laminar flow range. The main parameters of the VI were set as shown in Table 1.

After the key parameters were designed, we performed mesh-independence verification. In Figure 2, N indicates the number of grids. From Figure 2, it can be observed that the CE curve of the VI tended to be stable and consistent as the number of grids increased. When the number of grids reached $6\times 10^{4}$, the CE curves converged. This indicates that when the number of grids is larger than $6\times 10^{4}$, the results of the numerical model will not be affected by the number of grids, and that the results are reliable. Finally, we selected the grid number N equal to $6\times 10^{4}$ in consideration of the fact that the increase in grid number will increase the complexity of the model and computing time.

We then verified the irrelevance of the number of inlet particles. In Figure 3, M indicates the number of inlet particles released. From Figure 3, it can be observed that the CE curves tended to be consistent as the number of inlet particles increased. When M was larger than $5\times 10^{2}$, the CE curve hardly changed. We ultimately selected the number of entrance particles M equal to $5\times 10^{2}$ in consideration of the fact that an increase in the number of particles released increases operation time.

Ultimately, we obtained a preliminary numerical model of the virtual impactor with the convergence of the steady–state and transient solvers shown in Figure 4a,b.

## 3. Results and Discussion

### 3.1. Flow and Pressure Analysis

Figure 5 shows the velocity distribution of the VI at an inlet flow rate of 2.9 L/min and a splitting ratio of 10%. It was observed that the VI velocity field was symmetrically distributed and the major channel velocity was higher than the minor channel velocity. Since the splitting ratio was relatively small, the flow velocity of the minor flow channel was lower, which is conducive to the inertial separation of particles. When the flow velocity of the gas was faster, the smaller particles were more likely to follow the flow into the major flow channel; this result is consistent with our expectations.

Figure 6 shows the pressure distribution for the VI. It was observed that the pressure in the minor flow channel was considerably higher than that in the major flow channel, with a good pressure difference between the channels. The differential pressure was used to control the split ratio to maintain a ratio of 9:1, such that the flow distribution reduced wall loss.

Figure 7 shows the CE curves for different inlet flow conditions. It can be observed that the cutoff diameter decreased as the flow rate increased, which is consistent with the correlation reflected in the Stokes number equation. The results showed that for the present model configuration, the Stokes number of the cutoff diameter corresponded to around 0.7 for different flow rates. The cutoff diameter was 2.5 µm when the flow rate Q was equal to 3.7 L/min, and the final choice of inlet flow rate Q was 3.7 L/min when the corresponding ${\mathrm{Stk}}_{50}$ was 0.74.

### 3.2. Nozzle Inlet Length (Y) Analysis

Figure 8 shows the CE curves for different nozzle inlet lengths. It can be observed from the figure that the cutoff diameter oscillates between 2.51 µm and 2.54 µm when the nozzle inlet length varies between 4 mm and 12 mm. This shows that he nozzle inlet length has little effect on the performance of the virtual impactor. If a larger nozzle inlet length is selected, it may increase the probability of the particles hitting the wall in the entrance channel. This causes additional wall loss, which is detrimental to the performance of the virtual impactor. Meanwhile, the increase of the nozzle inlet length increases the size of the microfluidic chip, so we set the nozzle inlet length to 4 mm.

### 3.3. Splitting Ratio (r) Analysis

Figure 9 shows a graph of the CE with ${\mathrm{Stk}}_{50}$ as the horizontal coordinate for different splitting ratio conditions. It can be observed that the collection efficiency of 50% corresponding to ${\mathrm{Stk}}_{50}$ decreased as the splitting ratio increased. However, the steepness of the CE curve also decreased, the curve gradually leveled off, and the maximum collection efficiency decreased from 92% to 81%. This indicates that the WL of the VI also increased with the increase in the splitting ratio; thus, the chosen splitting ratio should not be too large, and the final splitting ratio chosen for this paper was 10%.

At the same time, we found that it was not possible to scale all the CE curves into one curve when changing the splitting ratio using ${\mathrm{Stk}}^{0.5}$. This is because the splitting ratio is not considered as a parameter in the Stokes number formula. This means that it is not possible to predict the cutoff diameter for different splitting ratios from the Stokes number. Thus, we proposed a modified Stokes number formula (${\mathrm{Stk}}_{m}$) that introduces the splitting ratio into the consideration of the Stoke number formula as shown in Equation (4).
(4)$${\mathrm{Stk}}_{m}=\mathrm{Stk}\times \left(a\times r^{b}\right)$$

Here, r is the splitting ratio, and a and b are the parameters associated with the VI model. After several numerical simulations, it was verified that a was equal to 1.35 and b was equal to 0.13. The modified Stokes number equation is shown below.
(5)$${\mathrm{Stk}}_{m}=\mathrm{Stk}\times \left(1.35\times r^{0.13}\right)$$

Figure 10 shows the scaling of our CE curves for different splitting ratio conditions using the modified Stokes number formula. From Figure 8, it can be observed that the corresponding ${\mathrm{Stk}}_{m}^{0.5}$ were located at 0.85 ± 0.05 for 50% collection efficiency, with the splitting ratio ranging from 0.05 to 0.25. We can therefore predict the cutoff diameter of the VI for different r values by using the modified ${\mathrm{Stk}}_{m}$.

### 3.4. Temperature (T) Analysis

In our previous study, we found that temperature has a non-negligible effect on the VI cutoff diameter due to the effect of fluid dynamic viscosity [16]. At temperatures less than 2000 K, the dynamic viscosity of the fluid can be calculated by the Sutherland formula, as shown in Equation (6) [25].
(6)$$\frac{\mu }{\mu_{0}}={\left(\frac{T}{288.15}\right)}^{1.5}\frac{288.15+B}{T+B}$$

Here, $\mu_{0}$ is equal to $1.7894\times 10^{-5}$ and B is a constant related to the gas type. From Equation (6), we can see that temperature and fluid dynamic viscosity are positively correlated, and according to the Stokes number, dynamic viscosity is positively correlated with the cutoff diameter. We can therefore reduce the cutoff diameter by decreasing the temperature. Figure 11 shows the plots of CE curves at different temperatures. As shown in Figure 11, there was a positive correlation between the temperature and the cutoff diameter, and the steepness of the CE curve was improved when the temperature decreased. When the temperature decreased from 293.15 K to 203.15 K, the cutoff diameter decreased from 2.5 µm to 2.05 µm, which was a decrease of about 18%. At the same time, the decrease in temperature weakened the particle Brownian motion phenomenon, which reduced the probability of particle wall loss due to Brownian motion and contributed to the improvement of VI performance.

### 3.5. Fluid Property Analysis

According to Equation (3), when the ${\mathrm{Stk}}_{50}$ of VI is determined, the fraction on the right side of the equation becomes fixed, as follows:(7)$${\mathrm{Stk}}_{50}=\frac{\rho_{p}d_{50}{\mathrm{QC}}_{c}}{9{\mu W}^{2}H}=0.59$$

In this case, if we reduce the dynamic viscosity μ and do not change other parameters such as the inlet flow rate and nozzle width, then the cutoff diameter $d_{50}$ is reduced. Thus, we can reduce the cutoff diameter by reducing the temperature or changing the fluid properties. We used CO_2_ as the fluid medium for VI instead of common air, and CO_2_ was chosen because the dynamic viscosity of CO_2_ at room temperature and pressure is about 80% of that of air, i.e., CO_2_ itself possesses the characteristic of low dynamic viscosity, which can achieve the aim of reducing the dynamic viscosity and therefore reducing the cutoff diameter. Another reason for choosing CO_2_ was that in practical applications, gaseous CO_2_ formed by the sublimation of dry ice can be used as a fluid medium not only to obtain the low dynamic viscosity of CO_2_ itself but also to reduce the temperature by using the cryogenic environment formed by its sublimation. This method combines the advantages of a cryogenic environment and CO_2_ gas to jointly reduce the VI cutoff diameter, which is superior to existing common methods.

Figure 12 and Figure 13 show the velocity and pressure distributions of the VI when CO_2_ was used as a fluid. From Figure 12 and Figure 13, it can be observed that the velocity field of the VI did not change when CO_2_ was used as a fluid. The velocity was still symmetrically distributed, and the flow velocity in the major flow channel was higher than that in the minor flow channel while the pressure on the minor flow channel of the VI increased, which was convenient for controlling the splitting ratio to maintain the low wall loss range.

To verify that the gaseous CO_2_ formed by dry ice sublimation has sufficient cooling capacity, we performed numerical simulations of its cooling effect via finite element analysis. The initial temperature inside the VI was set to 293.15 K, and the temperature of the incoming CO_2_ was the critical boiling point of dry ice sublimation at 195.15 K. The transient and steady-state solvers were used to solve for the laminar flow and fluid heat transfer, respectively. The temperature distribution inside the VI is shown in Figure 12. From Figure 14a, it can be observed that the ambient temperature of the transient solver inside the VI at 1 s stabilized between 194.6 K and 195 K (where the black arrows are velocity flow lines), which means that the temperature inside the VI decreased very rapidly, thus satisfying the cryogenic environment required. From Figure 14b, it can be observed that the VI state reached stability when the internal temperature was between 194.65 K and 195.08 K. The cooling effect was good and could provide a stable cryogenic environment for the VI. Combining the results of the transient and steady-state solutions, it can be observed that the cooling capacity of CO_2_ made by the sublimation of dry ice met the cryogenic environment required, and the results of the transient solver were basically consistent with the steady state when the time reached 1 s. Considering the efficiency of dry ice sublimation and the possible temperature loss in the actual environment, the numerical simulation temperature of 203.15 K was ultimately chosen.

Figure 15 shows the CE curves when either CO_2_ or air was used as the fluid. As can be seen in Figure 15, the cutoff diameter of VI when CO_2_ was used as the fluid was smaller than that when air was used as the fluid. The cutoff diameter was reduced from 2.5 µm to 2.15 µm, which was a 14% reduction. This is because the kinetic viscosity of CO_2_ is lower than that of air, which has the effect of reducing the kinetic viscosity, thus reducing the cutoff diameter.

### 3.6. Result of the Simulation

As mentioned earlier, we can reduce the cutoff diameter by using gaseous CO_2_ formed by dry ice sublimation as a fluid while obtaining the advantages of the cryogenic environment and low dynamic viscosity of CO_2_. We finally developed a CO_2_-driven cryogenic microfluidic chip model and proposed a new method to reduce the cutoff diameter. The CE curve of the obtained new virtual impactor is shown in Figure 16. The sufficient steepness of the CE curve indicates the good particle separation performance of the new CO_2_-driven microfluidic chip. The final cutoff diameter was reduced from 2.5 µm to 1.85 µm. This is a 26% reduction, which is superior to the existing conventional methods. Table 2 presents a comparison of this method with some other methods and effects, which shows that the method is remarkable and provides new direction for subsequent VI studies.

## 4. Conclusions

In this paper, we propose a new CO_2_-driven cryogenic virtual impactor model and explored the effects of the main structural parameters of the virtual impactor on performance through numerical simulations. We modified the Stokes number equation according to the effect of the splitting ratio so that the modified Stokes number could predict the cutoff diameter under different splitting ratios. Based on the relationship between temperature, fluid properties and fluid dynamic viscosity, we used cryogenic CO_2_ formed by the sublimation of dry ice instead of air as the fluid, which not only meant that the low kinetic viscosity of CO_2_ itself could be obtained, but the cryogenic environment formed by sublimation could also be obtained. The results show that temperature and fluid properties have a noticeable effect on virtual impactor performance. The virtual impactor with a cutoff diameter of 2.5 µm reduced the internal temperature of the VI from 293.15 K to between 194 K and 196 K after changing the fluid from air to cryogenic CO_2_, creating a stable cryogenic temperature field. The cutoff diameter of the VI was also reduced by 26%, which was superior to other methods of reducing the cutoff diameter in the same prototype configuration. We finally obtained a virtual impactor with a cutoff diameter of approximately 1.85 um and a good steepness, as shown in the CE curve. When the chip enters the actual preparation process, 3D printing technology can be used to make the chip prototype to ensure good sealing and precision. It is important to note that the pressure drop at the entrance stage should be considered to select the material for 3D printing. The preparation process and performance of the chip will be reported in detail in a subsequent paper. After the chip has been prepared, it can be applied to the field of particle mass concentration measurement. Considering its separation effect, the aerosol particles are first separated by particle size. Particle samples in the target particle size segment are collected and their mass concentration is measured using SAW sensors, QCM sensors or photoelectric sensors. Additionally, it can be integrated into a removable device and its operation can be controlled by programming. In future work, we intend to use both sublimation-formed cryogenic CO_2_ and sheath gas in the virtual impactor, which has the benefit of combining the advantages of cryogenics and sheath gas to reduce both the cutoff diameter and wall loss.

## Figures and Tables

**Figure 1 micromachines-14-00183-f001:**
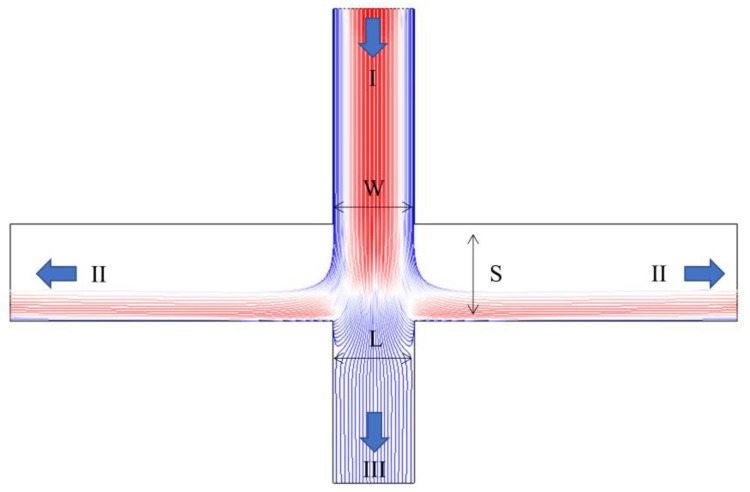
Particle streamline distribution of VI.

**Figure 2 micromachines-14-00183-f002:**
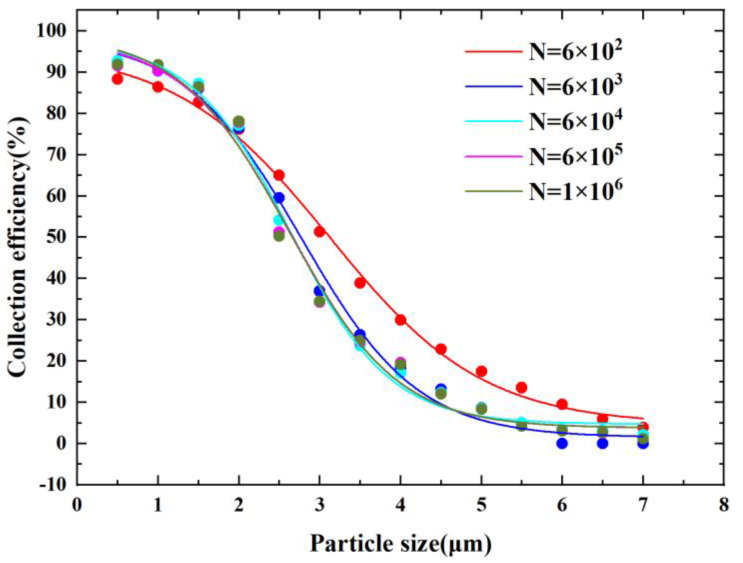
Grid irrelevance verification.

**Figure 3 micromachines-14-00183-f003:**
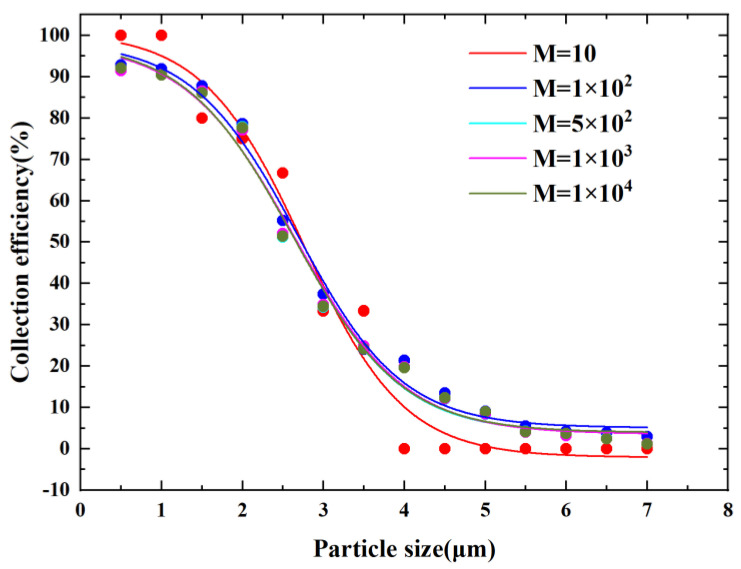
Particle number independence verification.

**Figure 4 micromachines-14-00183-f004:**
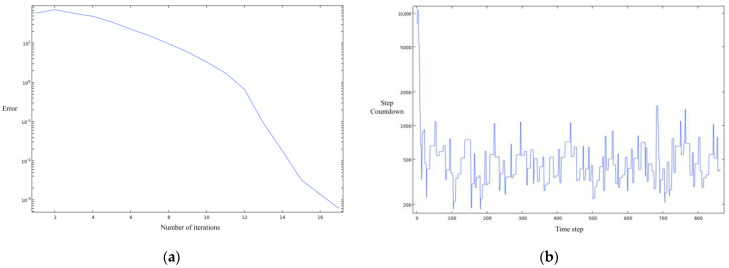
(**a**) steady–state solver; (**b**) transient solver.

**Figure 5 micromachines-14-00183-f005:**
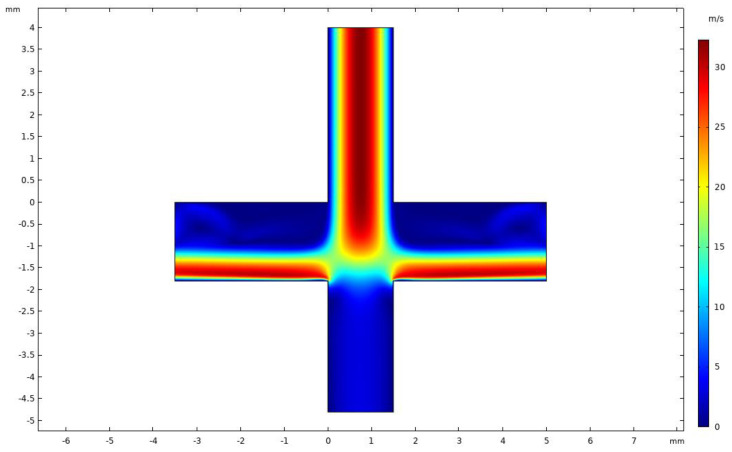
Velocity distribution in VI.

**Figure 6 micromachines-14-00183-f006:**
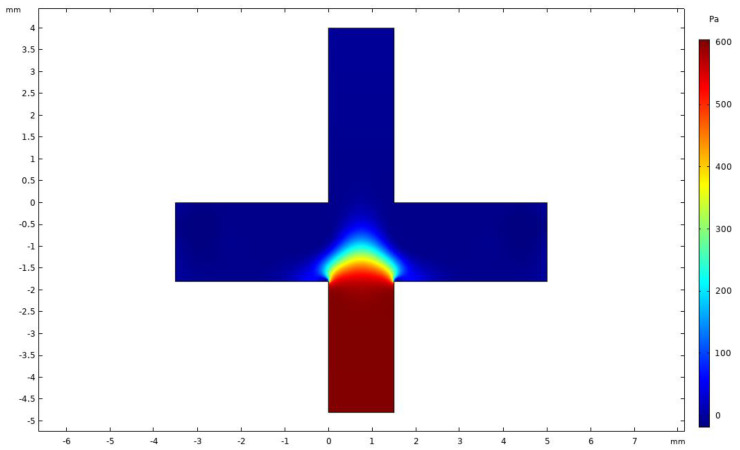
Pressure distribution in VI.

**Figure 7 micromachines-14-00183-f007:**
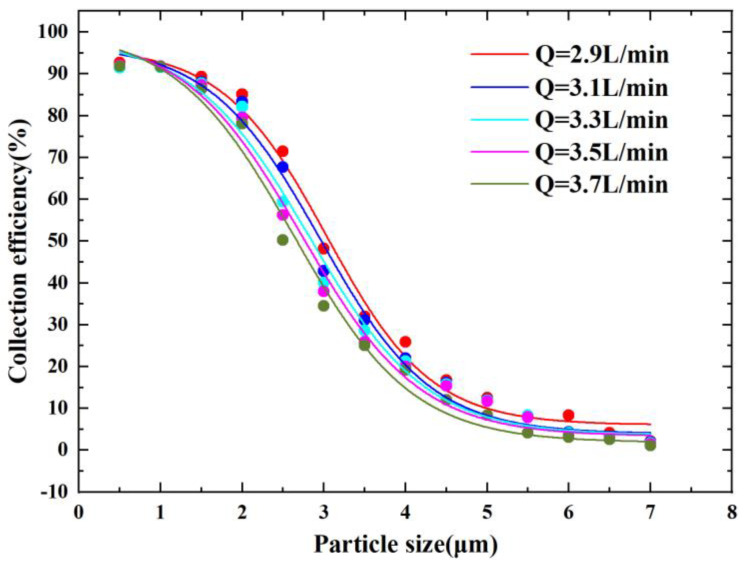
CE curves at different Q values.

**Figure 8 micromachines-14-00183-f008:**
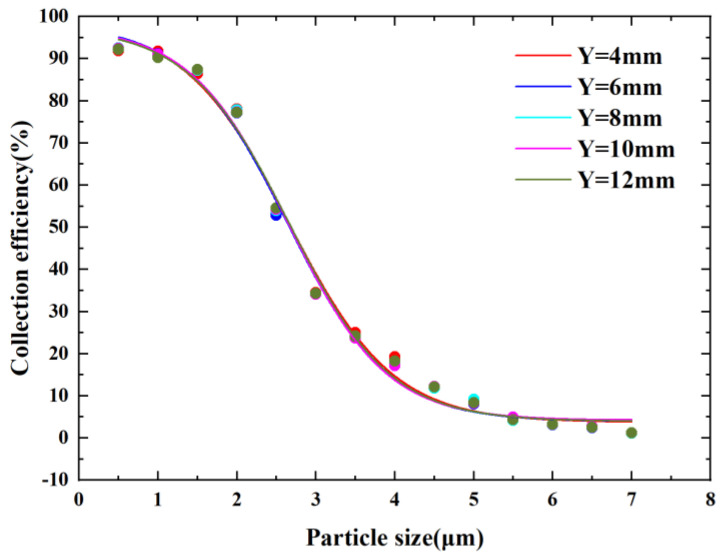
CE curves at different Y values.

**Figure 9 micromachines-14-00183-f009:**
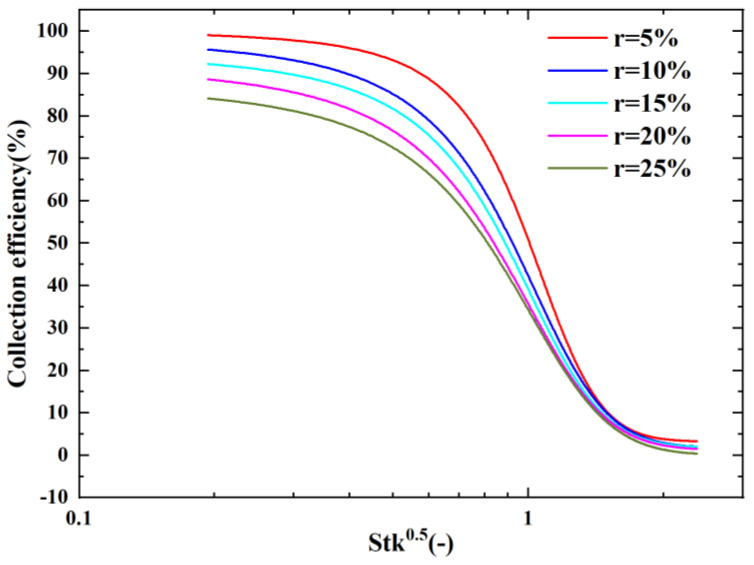
The effect of r on the CE curves using Stk^0.5^ as the absciss.

**Figure 10 micromachines-14-00183-f010:**
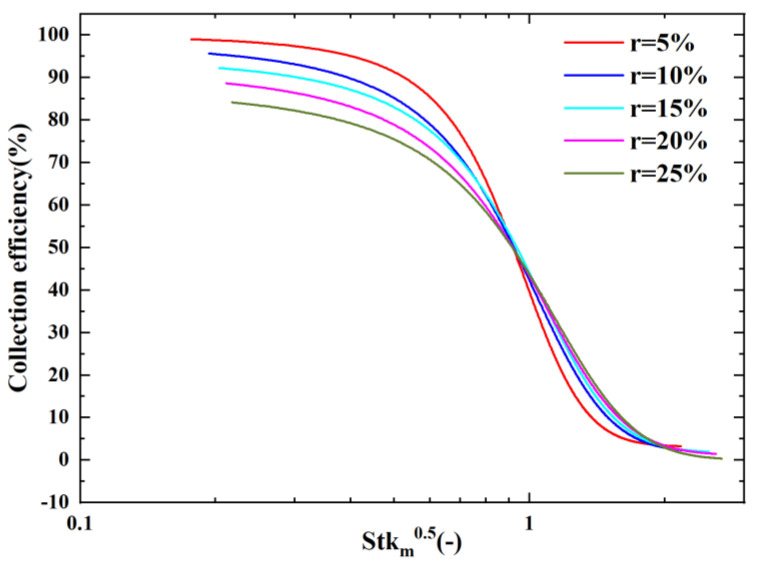
Effect of r on the CE curves using ${\mathrm{Stk}}_{m}^{0.5}$ as the absciss.

**Figure 11 micromachines-14-00183-f011:**
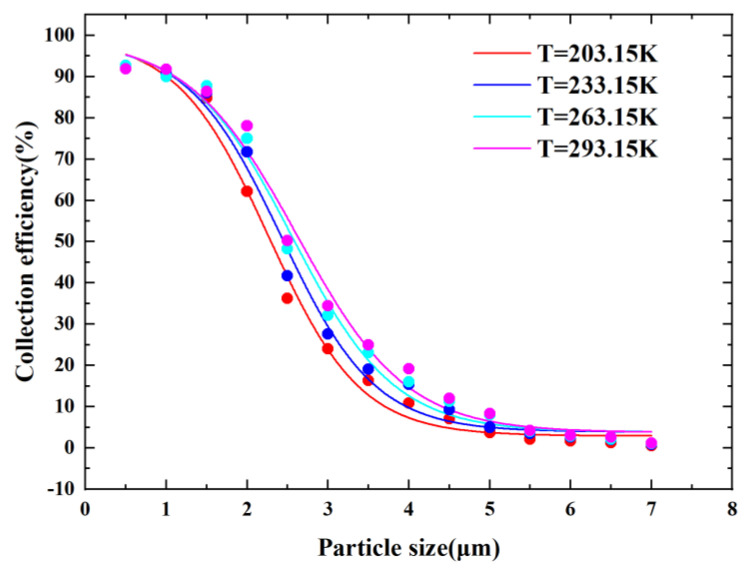
CE curves at different T value.

**Figure 12 micromachines-14-00183-f012:**
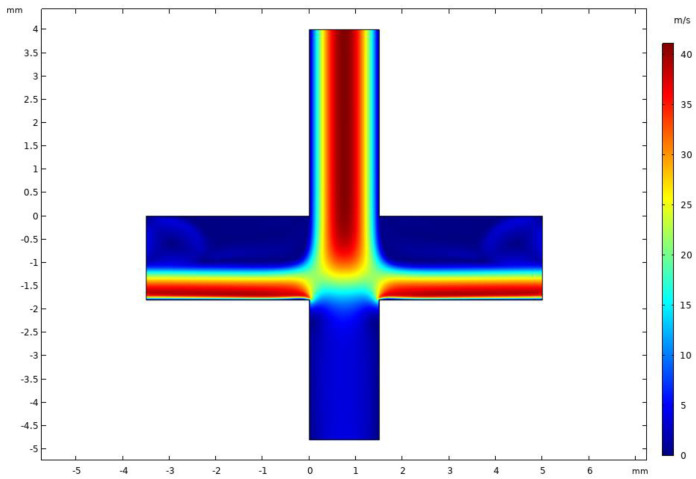
Velocity distribution of VI using CO_2_ as fluid.

**Figure 13 micromachines-14-00183-f013:**
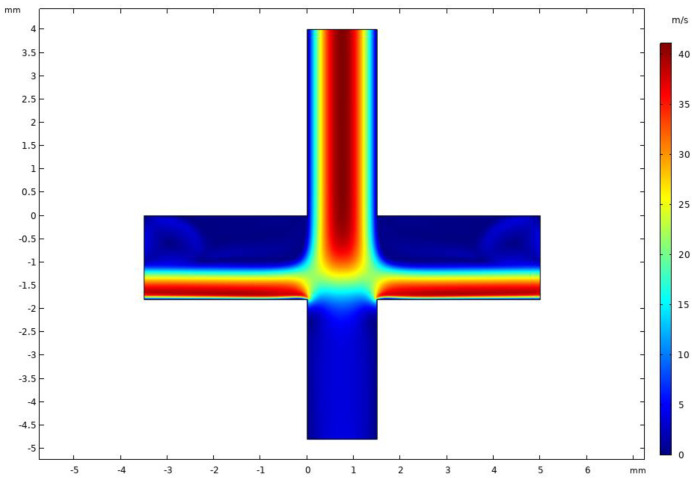
Pressure distribution of VI using CO_2_ as fluid.

**Figure 14 micromachines-14-00183-f014:**
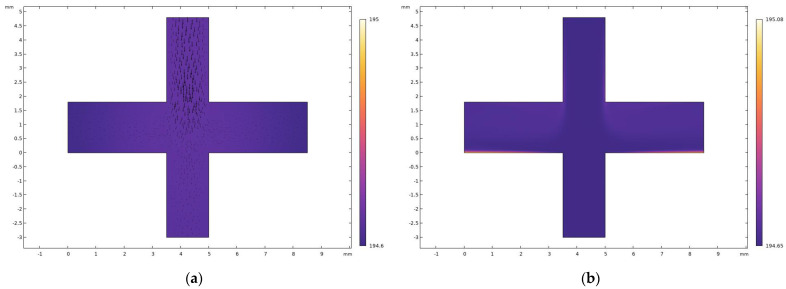
(**a**) Temperature distribution of VI at 1 s for the transient solver; (**b**) temperature distribution after stabilization of VI obtained by the steady-state solver.

**Figure 15 micromachines-14-00183-f015:**
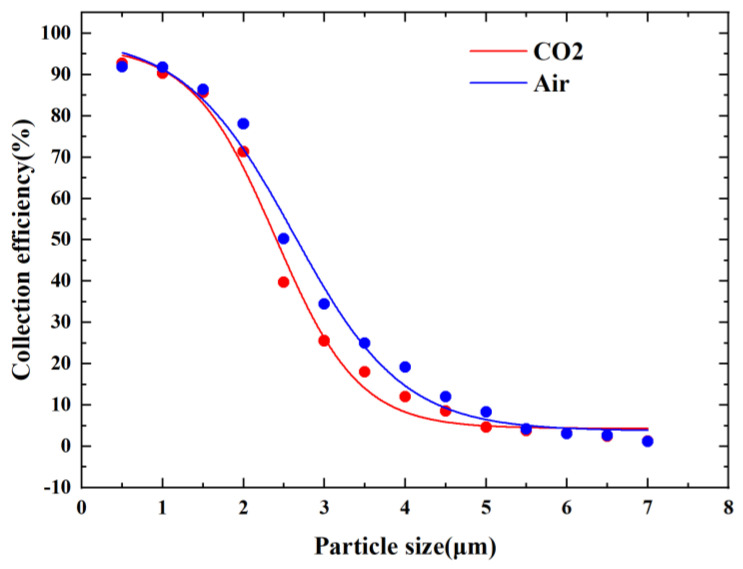
CE curves with different fluids.

**Figure 16 micromachines-14-00183-f016:**
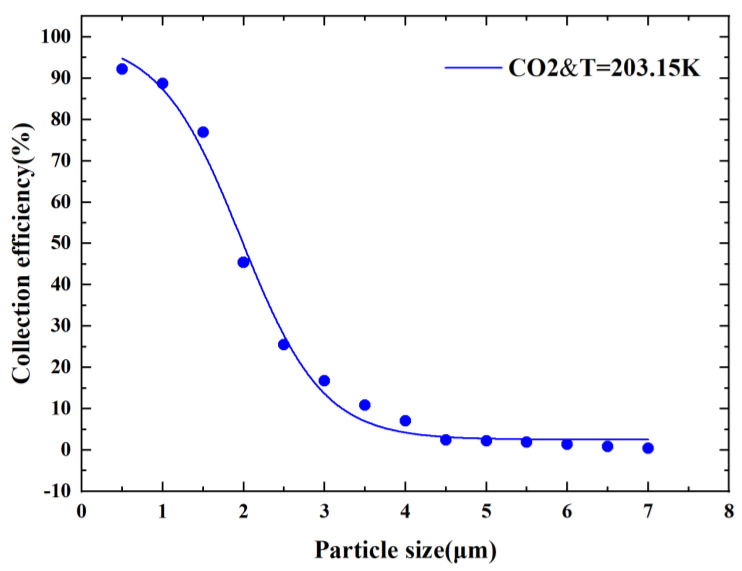
New virtual impactor collection efficiency curve.

**Table 1 micromachines-14-00183-t001:** Parameter design of new virtual impactor.

Parameter	Value	Unit
W (acceleration nozzle width)	1.5	mm
H (nozzle height)	1.5	mm
L (minor flow width)	1.5	mm
Y (nozzle inlet length)	4	mm
S (jet to the plate)	1.8	mm
r (splitting ratio)	10%	-
T (temperature)	203.15	K
Q (flow rate)	2.9	L/min
λ (Air molecule mean free range)	0.066	μm
$\rho_{p}$ (Particle density)	1000	Kg/m^3^
Fluid	CO_2_	-

**Table 2 micromachines-14-00183-t002:** Parameter design of new virtual impactor.

Property	Method	Reduction Effect
Ours	Dry ice	26%
Handol Lee et al. [11]	Orifice	16.7%
Se-Jin Yook et al. [14]	Horizontal inlet	16.8%

## Data Availability

Data sharing not applicable.

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
