# Peer review of "Designing a Microfluidic Chip Driven by Carbon Dioxide for Separation and Detection of Particulate Matter"

_micromachines, 2023, doi:10.3390/mi14010183_

Round 1
Reviewer 1 Report
Comments to the Author:
In this manuscript, the authors reported a virtual impactor model with a cutoff diameter of 1.85um based on numerical simulations by investigating many key parameters. The results indicate that temperature and fluid properties own a noticeable effect on the virtual impactor performance. The cut particle size was reduced from 2.5 μm to 1.85 μm after replacing conventional fluid air with CO2 formed by dry ice sublimation which is better than other existing methods for reducing the cutoff diameter. However, i have several concerns before this manuscript can be accepted. Therefore, in its current form, revisions are needed.
1. The design of key parameters such as, the nozzle inlet width, the nozzle inlet length and the shape of the nozzle, is not well presented. The authors need to make some discussions or conduct some simulations to optimize them, since these parameters will affect the cutoff diameters and wall loss of virtual impactors.
2. How did the authors ensure the accuracy of the simulation?
3. Comments on language:
- For example,
The virtual impactor with a cutoff diameter of 2.5 um reduces (Line 367);
Author Response
Please see the attachment, thanks.

Reviewer 2 Report
I have thoroughly reviewed the manuscript entitled "CO2 driven microfluidic chip design and analysis: For PM separation and detection". I think that the current manuscript is an incomplete manuscript for publication because the formation of this manuscript is inadequate and detailed explanation is not sufficiently provided. Taking into account the quality of work and scope of the journal, I would recommend the reject according to the following comments.
Minor comments:
# 1. Experimental results of this work are interesting, although the innovative points of this paper did not demonstrated clearly in the introduction part and results and discussions is not clearly presented.
# 2. I suggest that the authors summarize the specific characteristics of the virtual impactor in the form of a Table. Current version of introduction without the innovation point and importance of this study is not acceptable. My feeling is that the authors need to study a few examples of the numerical simulation reported in “high impact journals” before to resubmit the article.
# 3. Reference: Authors should be modified and unified style. Please read carefully the author guideline in this journal. Current manuscript is not acceptable.
# 4. The authors should improve their paper-writing with correct grammar and scientific scope.
Author Response
Please see the attachment, thanks.

Reviewer 3 Report
The following is my comments:
(1) The subscript of 2 in the the writing of carbon dioxide should be correctly shown across the whole manuscript, including the title.
(2) This centense is too long. It should be revised: "In this paper, we propose a new CO2-driven cryogenic virtual impactor model and
explore the effects of the main structural parameters of the virtual impactor on the perfor-
mance through numerical simulations, and modify the Stokes number equation according
to the effect of the splitting ratio, so that the modified Stokes number can predict the cutoff
diameter under different splitting ratios."
(3) The specific application of this work should be discussed in the end of the discussion section.
(4) The citation works are generally too old. Not quite sure whether they are up to date.
(5) The experimental fabrication of this device may be proposed or discussed in the end of the discussion part.
(6) The title is not good. It can be some thing like : " Designing a microfluidic chip driven by carbon dioxide for separation and detection of particulate matter".
Author Response
Please see the attachment, thanks.

Round 2
Reviewer 2 Report
The manuscript was corrected to the sufficient level for Micromachines